# The Preparation of FeCo/ZnO Composites and Enhancement of Microwave Absorbing Property by Two-Step Method

**DOI:** 10.3390/ma12193259

**Published:** 2019-10-06

**Authors:** Ka Gao, Junliang Zhao, Zhongyi Bai, Wenzheng Song, Rui Zhang

**Affiliations:** 1School of Materials Science and Engineering, Zhengzhou University of Aeronautics, Zhengzhou 450015, China; 201515020211@zut.edu.cn (J.Z.); zhybai@zua.edu.cn (Z.B.); wzsong@zua.edu.cn (W.S.); zhangray@zua.edu.cn (R.Z.); 2Henan Key Laboratory of Aeronautical Material and Application Technology, Zhengzhou 450015, China; 3School of Materials Science and Engineering, Zhengzhou University, Zhengzhou 450001, China

**Keywords:** two-step method, FeCo/ZnO composite, microstructure, absorption property

## Abstract

In this paper, the flower-like FeCo/ZnO composites were successfully firstly prepared by a two-step method, and their microstructures and microwave absorbing properties were characterized. The results show that with an increase of temperature, the content of ZnO loaded on a FeCo/ZnO composite surface was increased. The optimal reflection loss (RL) value can be reached around −53.81 dB at 9.8 GHz, which is obviously superior to results of previous studies and reports, and its effective bandwidth (RL < −10 dB) is 3.8 GHz in the frequency range of 8.7–11.8 GHz with a matching thickness of 1.9 mm. We considered that a large number of lamellar and rod-like ZnO loaded on nano-FeCo single-phase solid solution by two-step method can significantly improve the electromagnetic wave absorption properties.

## 1. Introduction

At present, microwave absorbing materials with lightweight electromagnetic are widely used in people’s livelihoods and military needs, and its property enhancement has also become a hot spot of domestic and international research [1,2,3,4,5]. Previous studies have found that the intrinsic electromagnetic wave absorption mechanisms of traditional single-phase ferrite [6], magnetic metal [7], C-type absorbing [8], semiconductor [9] and macromolecule absorbing materials [10] are mainly their own single dielectric loss and magnetic loss, which can no longer meet the need for an excellent absorbing property. Moreover, traditional single-phase wave absorbing materials has both disadvantages, such as easy oxidation of magnetic metals, and a high absorption band of semiconductor material [11]. Therefore, in recent years, in order to improve the absorption disadvantages of single magnetic metal and semiconductor, the microstructure and properties of traditional absorbing materials have been optimized by various method [7,8,9,10,11,12,13]. For example, using tunable chemical composition and high-energy ball milling method, Liu et al. [14] and Shang et al. [15] have optimized the disadvantage of easy oxidation of magnetic metals. Wen et al. [16] has prepared C/(Fe, Co, Ni) composite nano-powders by compounding multi-walled carbon nanotubes with magnetic metals, which exhibited electromagnetic wave absorption properties in the S-band (2–4 GHz). 

So far, the preparation methods of absorbing materials mainly include the chemical hydrothermal reduction method [11] and the high-energy ball milling method [15]. Among them, the chemical reduction is the most simple and inexpensive method. For example, Lv et al. [17] has obtained nano-FeCo/ZnO composites at 150 °C for 12 hours by traditional one-step method. However, because of less ZnO loaded on the surface of FeCo, this material only exhibited the optimal reflection loss of −31 dB with a 5.5 GHz effective frequency bandwidth. Due to the technical limitations of the one-step method, the hydrothermal method cannot optimize the absorbing properties of materials. Therefore, it is necessary to optimize the process on the basis of the traditional one-step method to improve the absorbing properties of materials.

Based on this, in this study we have firstly prepared the optimal microwave absorbing property of FeCo/ZnO composites by two-step method including the hydrothermal method and the template method. Its microstructure and absorbing property were characterized and analyzed. Furthermore, the effect of reaction temperature on the value of the minimum refection loss (RL) was investigated.

## 2. Experimental Procedures

### 2.1. Preparation of FeCo Single-Phase Solid Solution

In this study, FeCo single-phase solid solution was prepared by hydrothermal method. All chemical reagents were purchased from Kaitong Chemical Reagent Co. Ltd., Tianjin, China. The process for preparation of FeCo single-phase solid solution is as follows: Firstly, 1 mmol of FeSO_4_·7H_2_O and CoCl_2_·6H_2_O were dissolved in 60 mL deionized water under vigorous mechanical stirring for 0.5 h at room temperature. Then, 0.3 g C_6_H_5_Na_3_O_7_·2H_2_O, 3 g CH_3_COONa and 0.5 g NaOH were added in the above solution gradually and each maintained for 0.5 h. Next, 4 mL of N_2_H_4_·H_2_O (80 wt %) was added and then mixed slowly to make it even. After that, the above solutions were poured into the prepared reactor and sealed. Then the reactor was placed in the blast drying box which had been heated up to 150 °C in advance and maintained for 15 h. The substances produced by the reaction were collected by external magnets, then washed by distilled water and ethanol several times. Finally, the powder was collected and dried in vacuum for subsequent experiments. 

### 2.2. Preparation of FeCo/ZnO Composites

FeCo/ZnO composites with excellent properties were prepared by two-step method. Firstly, on the above basis, a FeCo single-phase solid solution was obtained. Then FeCo/ZnO composites were prepared by adding a Zn source using hydrothermal reaction. The experimental process was as follows: 0.1 g FeCo solid solution was added into a mixture of 20 mL absolute ethanol and 40 mL deionized water, which was maintained under vigorous mechanical stirring for 0.5 h at room temperature in order to make the mixture uniform. Then, 4 mmol dilute hydrochloric acid and 1 mmol of ZnCl_2_ were added in turn and each maintained for 0.5 h at room temperature. Afterward, the above solution was poured into the prepared reactor and sealed. Then the reactor was heated up to 120 °C, 150 °C, 180 °C and 200 °C in advance, respectively, and maintained for 15 h. Finally, the powder was collected, washed and dried in vacuum for subsequent experiments. The schematic of the synthetic progress for the FeCo/ZnO composites is shown in Figure 1.

### 2.3. Characterization

Microstructure and micro-area composition of the samples were analyzed by scanning electron microscopy (SEM, JSM-7001F, JEOL Ltd., Tokyo, Japan) and energy dispersive spectrometer analysis (EDS), respectively. The phase compositions of FeCo solid solution and FeCo/ZnO composites were determined by X-ray diffraction (XRD, SmartLab, Rigaku, Tokyo, Japan). At the same time, to measure the microwave absorption property, the absorbents were comprised of the samples and paraffin with a weight ratio of 3:2 and were then pressed into cylinder with an inner diameter of 3.04 mm, an outer diameter of 7.00 mm, and a thickness of 2 mm for this study. The electromagnetic parameters were determined in the frequency range of 2–18 GHz by a vector network analyzer (Agilent N5244A, Thermo Fisher Scientific, Waltham, MA, USA) and reflection loss (RL) was calculated according to the transmission line theory.

## 3. Results and Discussions

### 3.1. Microstructure and Microwave Absorption Property Characterization of FeCo Single-Phase Solid Solution

Figure 2 shows the FeCo single-phase solid solution obtained in this work. It can be seen that after the reduction reaction, irregular spheres with a diameter of about 150 nm were formed after the precipitation of Fe^2+^, Co^2+^ [17,18]. The particles were uniform in size, about 100 nm, and uniform in dispersion. Fe^2+^ and Co^2+^ were reduced by chemical reaction and finally formed a single-phase solid solution to avoid the easy oxidation of the single metal. Moreover, the surface of FeCo particles was not smooth, which is conducive to the development of back-loaded ZnO as reaction template. The EDS results showed that the irregular sphere was FeCo solid solution because of the more uniform distribution of the two elements, which was further proved by XRD analysis in Figure 3. The diffraction results were only consistent with the diffraction of FeCo solid solution in sample A of FeCo solid solution material.

Figure 4a,b show the dielectric constant and permeability of FeCo single-phase solid solution, respectively. The dielectric constant ε’ was decreased with frequency increasing, while the value of ε” was increased in Figure 4a. The permeability μ’ and μ” were all decreased along the same trend with frequency increasing in Figure 4b. According to the dielectric constant and permeability, the tangent values of dielectric loss (tanδ_ε_ = ε″/ε′) and magnetic loss (tanδ_μ_ = μ″/μ′) of the FeCo solid solution were calculated and are shown in Figure 4c. The tangent value of magnetic loss was increased gradually, with the frequency increasing larger than that of dielectric loss from 2 to 13.4 GHz. After reaching the maximum of 0.384 at 13.4 GHz, its value began to decrease gradually. However, the tangent value of dielectric loss was only increased linearly. The above result indicated that the absorbing mechanism of FeCo single-phase solid solution was mainly depended on magnetic loss [19].

The absorption properties of FeCo single-phase solid solution were tested and shown in Figure 5a. As can be seen from this figure, the effective absorbing frequency of FeCo solid solution gradually was moved to lower values with the absorbing matching thickness increasing, and the reflection loss (RL) value was increased. In particular, the optimal reflection loss (RL) value is reached at −42.26 dB at 15.3 GHz, almost near the results [20]. The effective bandwidth (RL ˂ −10 dB) was 5.6 GHz with a matching thickness of 1.4 mm. This was due to the hydrothermal reduction process in which the magnetic Fe and Co were formed to FeCo single-phase solid solution. The multiple polarization of electromagnetic wave was produced to promote the improvement of its absorbing properties. 

### 3.2. Microstructure and Microwave Absorption Property Characterization of FeCo/ZnO Composites

The FeCo solid solution was firstly covered as a reaction template in two-step process. ZnO was grown as irregular lamellar and rod-like [21], and then deposited on the surface of FeCo solid solution with a length of about 1 μm. A large number of flower-like FeCo/ZnO were stacked together in clusters with sizes ranging from 600 nm to 2 μm, as shown in Figure 6. The load of lamellar and rod-like ZnO formed a discontinuous conducting network and easily generated vibration microcurrent under the action of external magnetic field, which could be transmitted in the discontinuous conducting network. It was conducive to electromagnetic attenuation, thereby improving its absorbing performance. At the same time, the FeCo/ZnO composites were tested by EDS and XRD, respectively, as shown in Figure 6 and Figure 3. The results showed that the materials obtained were composed of Fe, Co and ZnO, in which the contents of Fe and Co were less than that of the ZnO. This was mainly due to the fact that a large amount of ZnO in this paper was completely loaded on the surface of FeCo solid solution (Figure 6), which made the EDS energy spectrum test results show less amount. Only FeCo solid solution phase and ZnO phase were existed in the material obtained in this paper in Figure 3. The above result further verified that the composites were FeCo/ZnO composites. 

Figure 7 shows the dielectric constant and permeability of FeCo/ZnO composites. The changing trends of value ε’ and ε” are shown in Figure 7a. Unlike the FeCo single-phase solid solution, FeCo/ZnO composites had a higher dielectric constant and better dielectric loss ability. Their value μ’ exhibited a general downward trend in the range of 2–18 GHz in Figure 7b, while the curve of μ” had two troughs, with minimum values at 3 GHz and 15 GHz, and a maximum value at 10.5 GHz. Further analysis showed that the changing trend of the dielectric loss tangent (tanδ_ε_ = ε″/ε′) of FeCo/ZnO composites was gentle and decreased gradually as the frequency increased, as shown in Figure 7c. While the variation trend of the magnetic loss tangent (tanδ_μ_ = μ″/μ′) was the same, the fluctuation was relatively large. Moreover, the dielectric loss between 4.7 GHz and 6.3 GHz was less than the tangent value of dielectric loss, indicating that in this frequency range the dielectric loss was the main type of absorption and in other frequencies it was the magnetic loss. This multiple loss mechanism is conducive to the improvement of the absorbing properties of composite materials [21]. 

Figure 5b shows the reflective loss (RL) of the prepared FeCo/ZnO composite under different matching thickness. The effective absorbing frequency gradually moved to a lower frequency with the matching thickness increasing. In particular, the optimal reflection loss (RL) value can reach −53.81 dB at 9.8 GHz at 180 °C and effective bandwidth (RL< −10 dB) is 3.8 GHz in the frequency range of 8.7–11.8 GHz with a matching thickness of 1.9 mm. The above results were obviously better than those reported at present [17,21].

### 3.3. Microstructure and Microwave Absorption Properties of FeCo/ZnO Composites at Different Temperatures

As we know, the reaction temperature had an important effect on the value of the minimum refection loss and the effective frequency width. In this study the effect of temperature on the microstructure and microwave absorption properties of FeCo/ZnO composites was investigated. At a singular temperature, FeCo/ZnO composites were all composed of FeCo single-phase solution and loaded ZnO in Figure 8. However, the amount of lamellar ZnO was less and its size was small at 120 °C in Figure 8a. As temperature increased, the amount of lamellar or rod-like ZnO began to increase. At 180 °C, the content of lamellar ZnO was the highest (Figure 8b). However, excessive temperature caused the dissolution of ZnO at 200 °C, and the lamellar structure was decreased [17], which directly affected the growth of lamellar ZnO, as shown in Figure 8c.

With the temperature increased from 120 °C to 180 °C, the absorption property was firstly enhanced, then decreased at 200 °C, as shown in Table 1. When at 120 °C, the reflection loss (RL) value was −33.34 dB at 8.2 GHz and effective bandwidth (RL< −10 dB) was 2.5 GHz with a matching thickness of 2.0 mm. The absorption property was enhanced to the best at 180 °C, which is obviously higher than other reported results [17,21,22,23], as shown in Table 2. However, the reflection loss (RL) value began to decrease to −39.56 dB at 9.2 GHz at 200 °C.

The enhancement of microwave absorption property may be related to the multiple microwave absorption mechanism of FeCo/ZnO composite microstructure and loaded ZnO content in this study. It is known that multiple reflection and scattering phenomena occurred in the FeCo solid solution. The energy was changed from electromagnetic energy to heat energy, and ultimately dissipation loss occurred to promote the improvement of its absorbing properties. However, unlike the FeCo single-phase solid solution, a large number of lamellar and rod-like ZnO loaded on FeCo surface may have brought multiple absorbing mechanisms to FeCo/ZnO composites [18,19]. The ZnO lamellar and rod on FeCo surface acted as an antenna receiver to receive electromagnetic waves into the material and generated discontinuous microcurrent, which can more effectively dissipate electromagnetic waves into the material. In addition, the size of the FeCo solid solution was reduced, and the eddy current effect was restrained [17,22]. The complex dielectric constant of the FeCo/ZnO composite was adjusted by introducing a ZnO rod to improve the impedance matching to form the synergistic effect. Under the combined action, its absorbing property can be improved and was obviously superior to other methods, as shown in Table 2. 

## 4. Conclusions

The FeCo/ZnO composites were successfully obtained with flower-like morphology by two-step method. The effect of temperature on microstructures and absorbing properties of FeCo/ZnO composites was investigated. When the temperature increased, the content of ZnO loaded on FeCo/ZnO composite surface firstly increased and then decreased. This resulted in the absorbing properties of FeCo/ZnO composite also firstly being enhanced and then decreased. When at 180 °C, optimal reflection loss (RL) value can reach −53.81 dB at 9.8 GHz better than other research results. 

## Figures and Tables

**Figure 1 materials-12-03259-f001:**
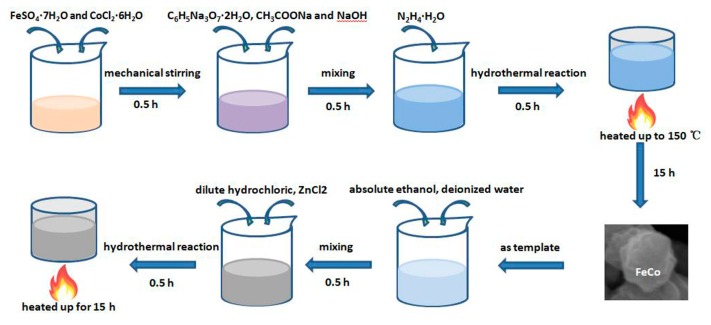
The schematic of the synthetic progress for the FeCo/ZnO composites.

**Figure 2 materials-12-03259-f002:**
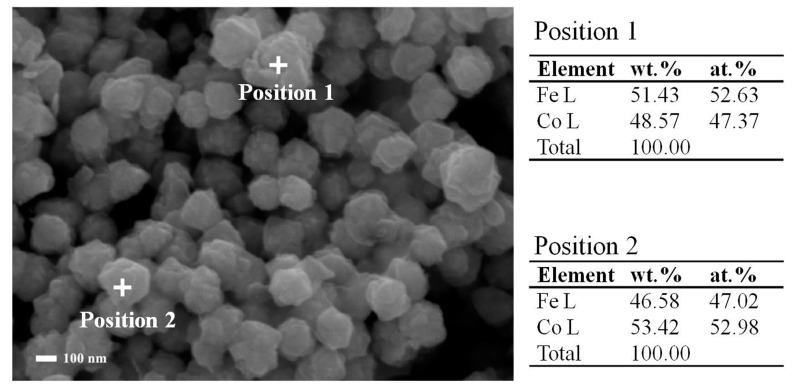
SEM and EDS of FeCo single-phase Solid Solution.

**Figure 3 materials-12-03259-f003:**
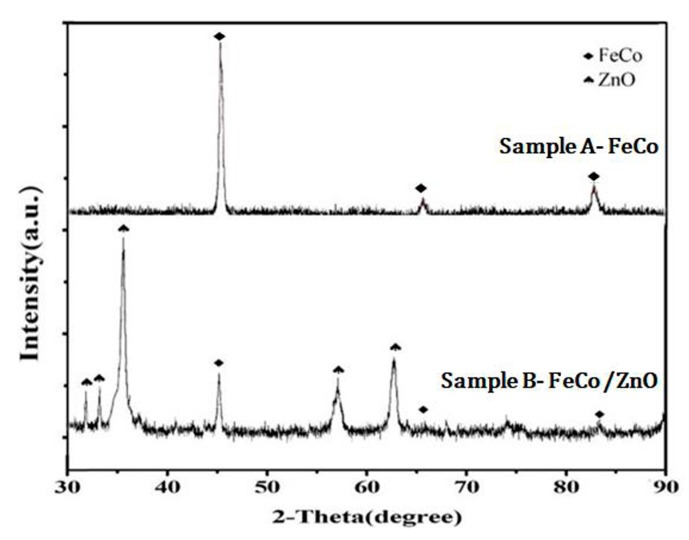
XRD diagram of FeCo single-phase solid solution and FeCo/ZnO composite.

**Figure 4 materials-12-03259-f004:**
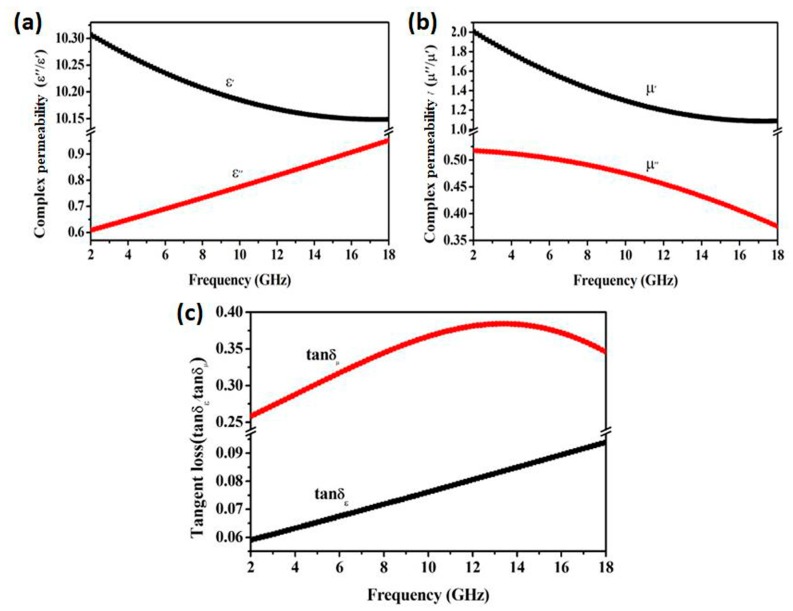
Dielectric constant (**a**), permeability (**b**) and loss factor (**c**) of FeCo single-phase solid solution sample.

**Figure 5 materials-12-03259-f005:**
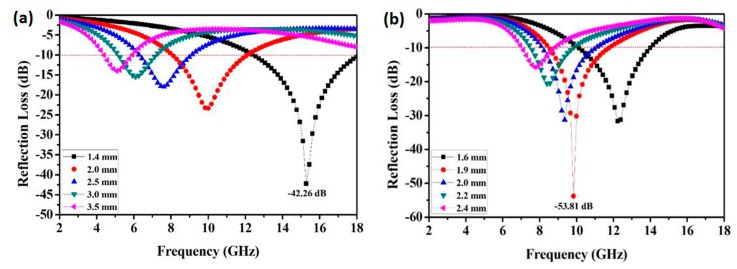
Reflection loss (RL) value of FeCo single-phase solid solution (**a**) and FeCo/ZnO composite (**b**).

**Figure 6 materials-12-03259-f006:**
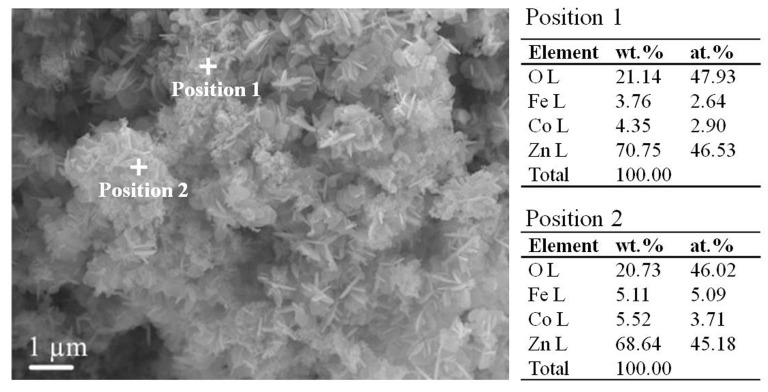
SEM and EDS of FeCo/ZnO composite.

**Figure 7 materials-12-03259-f007:**
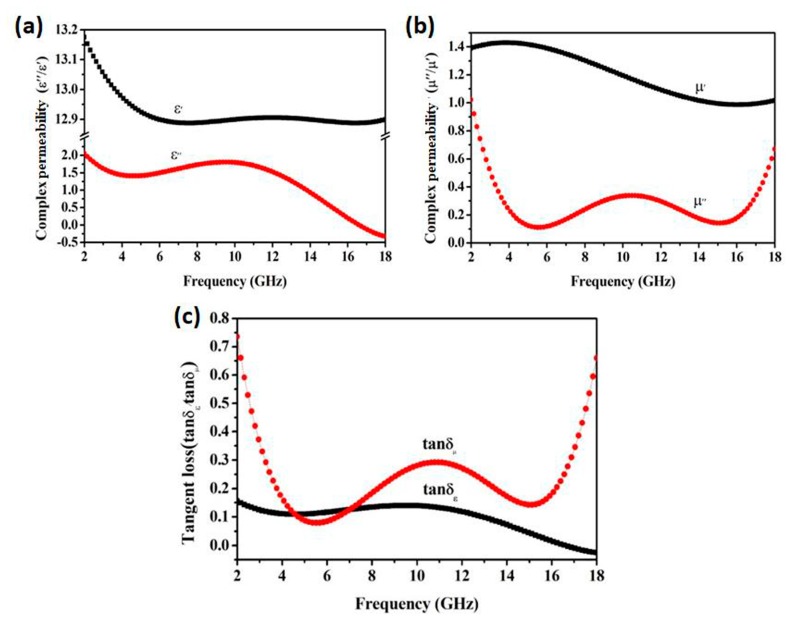
Dielectric constant (**a**), permeability (**b**) and loss factor (**c**) of FeCo/ZnO composite.

**Figure 8 materials-12-03259-f008:**
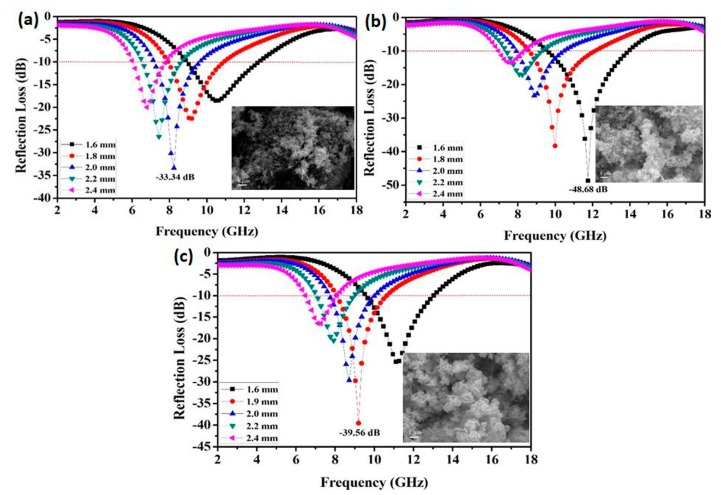
Reflection loss (RL) value and microstructure of FeCo/ZnO composite at different temperatures. (**a**) 120 °C; (**b**) 180 °C; (**c**) 200 °C.

**Table 1 materials-12-03259-t001:** Microwave absorption properties of FeCo/ZnO composites at different temperatures (120 °C; 150 °C; 180 °C and 200 °C).

Temperature	Integrated Thickness (mm)	Absorption Frequency (GHz)	Reflection Loss (dB)	Effective Bandwidth (GHz)
120 °C	2.0	8.2	−33.34	2.5
150 °C	1.6	11.8	−48.68	4.5
180 °C	1.9	9.8	−53.81	3.8
200 °C	1.9	9.2	−39.56	2.5

**Table 2 materials-12-03259-t002:** Optimal microwave absorption properties of FeCo and FeCo/ZnO composites reported in the different literatures.

Materials	Prepared Method	Absorption Frequency (GHz)	Reflection Loss (dB)
FeCo/ZnO	Our method	9.8	−53.81
FeCo	Our method	15.3	−42.26
FeCo/ZnO	One-step hydrothermal method [17]	5.5	−31
FeCo/ZnO	Liquid phase reduction method [21]	14.8	−34.8
FeCo	Liquid phase reduction method [22]	7.1	−22
Fe7Co3	Liquid–thermal reduction method [18]	14.3	−53.6
Fe40Co60	Mechanical alloying route method [23]	9.75	−11.3

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
