# Peer review of "The Preparation of FeCo/ZnO Composites and Enhancement of Microwave Absorbing Property by Two-Step Method"

_materials, 2019, doi:10.3390/ma12193259_

Round 1
Reviewer 1 Report
The topic of the manuscript entitled “The FeCo/ZnO composites prepared and microwave absorbing property enhanced by two-step method” falls within the scope of the journal Materials. The paper contains very interesting experimental results. It is of sufficient scientific interest and has originality in its technical content to merit publication. The authors have cited the relevant literature. Methods, interpretations of results and conclusions are correct and novel. In terms of content, the analysis does not raise any objections. The arrangement of work needs improvement. In my opinion, the manuscript is not suitable for publication in its present form.
Detailed comments are provided below.
The descriptions in Figure 4c (tan-delta) must be improved - it is difficult to read. Conclusions should be specified in points rather than in general descriptive form. The Reference list should be made in accordance with Instructions for Authors.Author Response
Please see the attachment.

Reviewer 2 Report
The conclusions should be extended. The are some small mistakes, in the text (for. example fig. 8).

Reviewer 3 Report
The work ineeds improvement so I would not like to recommend its publication in Materials in a current form of manuscript.
I would suggest a major revision for the authors to address the following concerns.
L12
…morphology and flower-like micro-nano FeCo/ZnO composites were successfully prepared by a two-step method,…
Insert “by a”
L26-27At present, lightweight electromagnetic wave absorbing materials have become the key demand
of people's livelihood and military at home and abroad….???
Rephrase
L 32-33…Moreover, traditional wave absorbing materials with single-phase has both disadvantages,…
I suggest replacing with single-phase materials. What single-phase materials? Composition? Or in general?
L 46…to enhance wave absorbing, this material was only exhibited the optimal reflection loss of -31 dB with….
L 59…All chemical reagents in analysis analytical grade were
L63… CH3COONa and 0.5 g NaOH were in turn added to the above solution and each maintained for…
L64…Next 4 mL of N2H4·H2O (80 wt.%) was added and mixing was mixed slowly to make it even…
L 66…And The substances produced by the reaction were collected by with external magnets and washed by with distilled water and ethanol for several times
L 75… which was maintained for 0.5 h in order to make the mixture uniform..
Maintained how? At a specific temperature? At specific mixing rot/ min?
L76
Then, 0.1 mol/L dilute hydrochloric acid with 4 mL and 1 mmol of ZnCl2 were in turn added to the above solution and each maintained for 0.5 h at room temperature…
Rephrase, I could not understand. HCl 0.1 M with 4 ml of what?
L103
Fe2+ and Co2+ had been formed a single-phase…
L 106…which was further proved
L107The diffraction results were consistent with the diffraction of FeCo solid solution, and no other unnecessary and disordered diffraction peaks had been found.
Rephrase. The affirmation is not correct because in Fig 3 are the XRD patterns for FeCo and FeCo/ZnO.
Fig 3- Poor quality of the figure. Note on the graphic each XRD pattern, for example
sample A - FeCo
sample B - FeCo/ZnO
With what CDB files from the Rigaku database have, you correlated your results?
L 110Figure 3. XRD diagram of FeCo signal-phase single-phase solid solution and FeCo/ZnO composite
Fig 4- Poor quality of the figure
L 116
…. were calculated and shown in Fig. 4(c).
L117
Its tangent value of magnetic loss was increased gradually with the frequency increasing larger than that of dielectric loss in the whole frequency range.
Rephrase.
L 118When It reached a maximum of 0.384 at 13.4 GHz, then gradually decreased.
But The tangent value of dielectric loss was all linear increased.
Rephrase.
L119The above result was indicated that the mechanism of…
L125
Therefore, the absorption properties of FeCo single-phase solid solution were tested and they are showed in Fig. 5(a).
L127
…effective absorbing frequency of FeCo solid solution gradually was moved to lower values
L144
The FeCo/ZnO composites were further characterized by EDS and XRD, respectively, as shown in Fig. 6 and Fig. 3.
The XRD analysis was already discussed and is not a further characterization.
L149
From the results of XRD in Fig. 3, only FeCo solid solution phase and ZnO phase was existed, which further verified that the composites obtained in this paper were FeCo/ZnO composites.
Rephrase.
Fig 7 - Poor quality of the figureL 175
As we know, the reaction temperature was had an important effect …
L 176So the effect of temperature on the microstructure and microwave absorption properties of FeCo/ZnO composites was investigated in this study.
Rephrase.
In this study was investigated the effect…..
Fig 8 - Poor quality of the figure L187Move the temperatures at L 186.
L 188
While With the temperature increased from 120 ℃ to180 ℃, its the absorption property was first enhanced firstly, then decreased at 200 ℃ as is shown in Table 1. When At 120 ℃ the reflection loss (RL) value was -33.34 dB at 8.2 GHz and effective bandwidth (RL< -10 dB) was 2.5 GHz with a matching thickness of 2.0 mm.
L 191And its absorption property was enhanced to the best at 180 ℃. However, the reflection loss (RL) value was began to decrease to -39.56 dB at 9.2 GHz at 200 ℃.
Rephrase.
Please present results comparatively. Reformulate the conclusions.(are presented to many results)
Author Response
Dear reviewer 3,Re: Manuscript No.: materials-589787
Thank you very much for your reply and help. Thanks a lot for the reviewers’ comments and their kind suggestions of our manuscript entitled “The FeCo/ZnO composites prepared and microwave absorbing property enhanced by two-step method”. These comments are all valuable and very helpful for revising and improving our paper, as well as the important guiding significance to our researches. We provide this letter point by point and explain the details of our revisions in the manuscript. In the revised manuscript, we make the revision in red. Besides, we have carefully checked the manuscript and the responses to the reviewers’ comments as follows.
Reviewer #3:
Comment: L12…morphology and flower-like micro-nano FeCo/ZnO composites were successfully prepared by a two-step method,…Insert “by a”
Reply: Thank you for your comment. "by the two-step method, " has been replaced by “by a two-step method,” in red shown in the revised manuscript.
Comment:L26-27At present, lightweight electromagnetic wave absorbing materials have become the key demand of people's livelihood and military at home and abroad….??? Rephrase
Reply: Thank you for your comment. This sentence has been rewritten in red shown in the revised manuscript.
Comment:L 32-33…Moreover, traditional wave absorbing materials with single-phase has both disadvantages,…I suggest replacing with single-phase materials. What single-phase materials? Composition? Or in general?
Reply: Thank you for your comment. Single-phase materials means traditional single-phase ferrite, magnetic metal, C-type absorbing, semiconductor, in which its phase composition is a phase, not two or more phases. According to your comment, " single-phase" has been replaced by “single-phase materials” in red shown in the revised manuscript.
Comment:L 46…to enhance wave absorbing, this material was only exhibited the optimal reflection loss of -31 dB with….
Reply: This sentence has been rewritten by “because of fewer ZnO loaded on the surface of FeCo, this material was only exhibited the optimal reflection loss of -31 dB with a 5.5 GHz effective frequency bandwidth.” in red shown in the revised manuscript.
Comment:L 59…All chemical reagents in analysis analytical grade were
Reply: This sentence has been rewritten by “All chemical reagents were” in red shown in the revised manuscript.
Comment:L63… CH3COONa and 0.5 g NaOH were in turn added to the above solution and each maintained for…
Reply: This sentence has been rewritten by “Then, 0.3 g C6H5Na3O7·2H2O, 3 g CH3COONa and 0.5 g NaOH were added in the above solution gradually and each maintained for 0.5 h.” in red shown in the revised manuscript.
Comment:L64…Next 4 mL of N2H4·H2O (80 wt.%) was added and mixing was mixed slowly to make it even…
Reply: This sentence has been rewritten by “Next 4 mL of N2H4·H2O (80 wt.%) was added and then mixed slowly to make it even.” in red shown in the revised manuscript.
Comment:L 66…And The substances produced by the reaction were collected by with external magnets and washed by with distilled water and ethanol for several times
Reply: This sentence has been rewritten by “And The substances produced by the reaction were collected by with external magnets, and then washed by with distilled water and ethanol for several times” in red shown in the revised manuscript.
Comment:L 75… which was maintained for 0.5 h in order to make the mixture uniform..Maintained how? At a specific temperature? At specific mixing rot/ min?
Reply: Thank you for your comment. The experimental process is as follows: 0.1 g of the as-prepared FeCo solid solution was added into the mixture of 20 ml absolute ethanol and 40 ml deionized water, which was maintained under vigorous mechanical stirring for 0.5 h at room temperature in order to make the mixture uniform. So this sentence has been rewritten by “The experimental process is as follows: 0.1 g of the as-prepared FeCo solid solution was added into the mixture of 20 ml absolute ethanol and 40 ml deionized water, which was maintained under vigorous mechanical stirring for 0.5 h at room temperature in order to make the mixture uniform.” in red shown in the revised manuscript.
Comment:L76Then, 0.1 mol/L dilute hydrochloric acid with 4 mL and 1 mmol of ZnCl2 were in turn added to the above solution and each maintained for 0.5 h at room temperature…Rephrase, I could not understand. HCl 0.1 M with 4 ml of what?
Reply: Thank you for your comment. This sentence has been rewritten by “Then, 4 mmol dilute hydrochloric acid and 1 mmol of ZnCl2 were in turn added the above solution and each maintained for 0.5 h at room temperature.” in red shown in the revised manuscript.
Comment:L103Fe2+ and Co2+ had been formed a single-phase…
Reply: This sentence has been rewritten by “Fe2+ and Co2+ had been formed a single-phase solid solution” in red shown in the revised manuscript.
Comment:L 106…which was further proved
Reply: " which was further prove" has been rewritten by “which was further proved” in red shown in the revised manuscript.
Comment:L107The diffraction results were consistent with the diffraction of FeCo solid solution, and no other unnecessary and disordered diffraction peaks had been found. Rephrase. The affirmation is not correct because in Fig 3 are the XRD patterns for FeCo and FeCo/ZnO.Fig 3- Poor quality of the figure. Note on the graphic each XRD pattern, for examplesample A - FeCosample B - FeCo/ZnOWith what CDB files from the Rigaku database have, you correlated your results?
Reply: Thank you for your comment. According to your comment, this sentence has been rephrased by “The diffraction results were consistent with the diffraction of FeCo solid solution in sample A of FeCo solid solution material.” in red shown in the revised manuscript. Moreover, the Fig. 3 was redraw to add the graphic each XRD pattern, including sample A-FeCo and sample B-FeCo/ZnO. And the Fig.3 with higher quality has been uploaded again. In our work, we have used the X-ray diffraction standard diagram of FeCo and ZnO from the Rigaku database to correlate our results.
Comment:L 110Figure 3. XRD diagram of FeCo signal-phase single-phase solid solution and FeCo/ZnO compositeFig 4- Poor quality of the figure
Reply: Thank you for your comment. This sentence was rewritten in red shown in the revised manuscript. Moreover, the Fig. 4 was redraw and the Fig.4 with higher quality has been uploaded again.
Comment:L 116…. were calculated and shown in Fig. 4(c).
Reply: Thank you comment. " were calculated shown in Fig. 4(c)." has been replaced by “were calculated and shown in Fig. 4(c).” in red shown in the revised manuscript.
Comment:L117Its tangent value of magnetic loss was increased gradually with the frequency increasing larger than that of dielectric loss in the whole frequency range. Rephrase.
Reply: "Its tangent value of magnetic loss was increased gradually with the frequency increasing larger than that of dielectric loss in the whole frequency range. " has been replaced by “Its tangent value of magnetic loss was increased gradually with the frequency increasing larger than that of dielectric loss from 2 to 13.4 GHz.” in red shown in the revised manuscript.
Comment:L 118When It reached a maximum of 0.384 at 13.4 GHz, then gradually decreased. But The tangent value of dielectric loss was all linear increased. Rephrase.
Reply: " When It reached a maximum of 0.384 at 13.4 GHz, then gradually decreased. But The tangent value of dielectric loss was all linear increased." has been replaced by “After reached a maximum of 0.384 at 13.4 GHz, its value was began to decrease gradually. But the tangent value of dielectric loss was only linear increasing.” in red shown in the revised manuscript.
Comment:L119The above result was indicated that the mechanism of…
Reply: Thank you for your comment. " The above result was indicated that the mechanism of absorbing property of FeCo single-phase solid solution was mainly depended on magnetic loss" has been replaced by “The above result was indicated that the absorbing mechanism of FeCo single-phase solid solution was mainly depended on magnetic loss” in red shown in the revised manuscript.
Comment:L125Therefore, the absorption properties of FeCo single-phase solid solution were tested and they are showed in Fig. 5(a).
Reply: This sentence has been revised in red shown in the revised manuscript.
Comment:L127…effective absorbing frequency of FeCo solid solution gradually was moved to lower values
Reply: Thank you for your comment. This sentence has been revised in red shown in the revised manuscript.
Comment:L144The FeCo/ZnO composites were further characterized by EDS and XRD, respectively, as shown in Fig. 6 and Fig. 3. The XRD analysis was already discussed and is not a further characterization.
Reply: Thank you for your comment. "The FeCo/ZnO composites were further characterized by EDS and XRD, respectively, as shown in Fig. 6 and Fig. 3." has been replaced by “The FeCo/ZnO composites were tested by EDS and XRD, respectively, as shown in Fig. 6 and Fig. 3.” in red shown in the revised manuscript.
Comment:L149From the results of XRD in Fig. 3, only FeCo solid solution phase and ZnO phase was existed, which further verified that the composites obtained in this paper were FeCo/ZnO composites. Rephrase.Fig 7 - Poor quality of the figure
Reply: Thank you for your comment. This sentence was rewritten as "Only FeCo solid solution phase and ZnO phase was existed in Fig. 3. The above result was further verified that the composites obtained in this paper were FeCo/ZnO composites." in red shown in the revised manuscript. Moreover, the Fig. 7 with higher quality was redraw and uploaded again.
Comment:L 175As we know, the reaction temperature was had an important effect …
Reply: Thank you for your comment. This sentence has been rewritten in red shown in the revised manuscript.
Comment:L 176So the effect of temperature on the microstructure and microwave absorption properties of FeCo/ZnO composites was investigated in this study. Rephrase.In this study was investigated the effect…..Fig 8 - Poor quality of the figure L187
Reply: Thank you for your comment. This sentence has been replaced by “In this study was investigated the effect of temperature on the microstructure and microwave absorption properties of FeCo/ZnO composites.” in red shown in the revised manuscript. Moreover, the Fig. 8 with higher quality was redraw and uploaded again.
Comment:L 188While With the temperature increased from 120 ℃ to180 ℃, its the absorption property was first enhanced firstly, then decreased at 200 ℃ as is shown in Table 1. When At 120 ℃ the reflection loss (RL) value was -33.34 dB at 8.2 GHz and effective bandwidth (RL< -10 dB) was 2.5 GHz with a matching thickness of 2.0 mm.
Reply: This sentence has been rewritten in red shown in the revised manuscript.
Comment:L 191And its absorption property was enhanced to the best at 180 ℃. However, the reflection loss (RL) value was began to decrease to -39.56 dB at 9.2 GHz at 200 ℃.
Reply: Thank you comment. "by the two-step method, " has been replaced by “by a two-step method,” in red shown in the revised manuscript.
Once again, thank you very much for your comments and suggestions. We hope the revisions and responses are sufficient and the resubmitted manuscript is suitable for publication. We shall look forward to hearing from you at your earliest convenience.
Have a good day.
Yours sincerely,
K.GAO, Junliang Zhao, Zhongyi Bai, Wenzheng Song and Rui Zhang
Zhengzhou University of Aeronautics,
Zhengzhou 450015,
R. China
Corresponding authors. E-mail: gaoka9222005@163.com (K. Gao)
Reviewer 4 Report
The authors present a 2-step method for fabricating FeCo/ZnO composites for microwave absorption property. While the topic is interesting, the study lacks clarity and relevant literature support. The following comments are appended for the authors to improve their manuscript.
Please rephrase the fragment in P1L42-P2L50: there is a need for more clarity. What is the difference between one-step (P1L45) and traditional one-step method (P2L49)? Please introduce the template method. Explain “Then, 0.1 mol/L dilute hydrochloric acid with 4 ml and 1 mmol of ZnCl2 were in turn added the above solution and each maintained for 5 h at room temperature” The formation mechanism for FeCo particles is not clear. Please rephrase the fragment P3L103-108. How do the more regular particles form with respect to the irregular-shaped ones? The identification of XRD peaks in fig 3 FeCo is wrong. Please correct. Please comment figure 3 with findings on the composite. Figures 2-8 are presented without any discussion. Please rework. The title needs grammar correction. Please correct: ‘improve the absorption disadvantages’ Please rephrase and correct P1L38-39: has changed the disadvantage Please correct grammar in P1L39-40. Please correct grammar in P2L46.Author Response
Dear reviewer 4,Re: Manuscript No.: materials-589787
Thank you very much for your reply and help. Thanks a lot for the reviewers’ comments and their kind suggestions of our manuscript entitled “The FeCo/ZnO composites prepared and microwave absorbing property enhanced by two-step method”. These comments are all valuable and very helpful for revising and improving our paper, as well as the important guiding significance to our researches. We provide this letter point by point and explain the details of our revisions in the manuscript. In the revised manuscript, we make the revision in red. Besides, we have carefully checked the manuscript and the responses to the reviewers’ comments as follows.
Reviewer #4:
Comment: Please rephrase the fragment in P1L42-P2L50: there is a need for more clarity. What is the difference between one-step (P1L45) and traditional one-step method (P2L49)? Please introduce the template method. Explain “Then, 0.1 mol/L dilute hydrochloric acid with 4 ml and 1 mmol of ZnCl2 were in turn added the above solution and each maintained for 5 h at room temperature”. The formation mechanism for FeCo particles is not clear. Please rephrase the fragment P3L103-108. How do the more regular particles form with respect to the irregular-shaped ones? The identification of XRD peaks in fig 3 FeCo is wrong. Please correct. Please comment figure 3 with findings on the composite. Figures 2-8 are presented without any discussion. Please rework. The title needs grammar correction. Please correct: ‘improve the absorption disadvantages’ Please rephrase and correct P1L38-39: has changed the disadvantage Please correct grammar in P1L39-40. Please correct grammar in P2L46.
Reply: Thank you for your useful comment. Firstly, we are sorry for the badly writing result in reading difficultly. In order to better enable readers to understand this description, a re-writing has been revised and added in red shown in P1L42-P2L50 of the revised manuscript. In this paper, there has no difference between the one-step (P1L45) intruded and traditional one-step method (P2L49). They are the same. So we have revised the one-step (P1L45) to traditional one-step method, which is in red shown in P1L45 of the revised manuscript.Template method is a synthetic method which takes template as the main configuration to control, influence and modify the morphology of materials, control the size and then determine the properties of materials.The new explanation was revised to replace "Then, 0.1 mol/L dilute hydrochloric acid with 4 ml and 1 mmol of ZnCl2 were in turn added the above solution and each maintained for 5 h at room temperature", which is in red shown in part of 2.2 in the revised manuscript.The formation mechanism for FeCo particles has been rephrased in red shown in P3L103-108 in the revised manuscript.Due to the addition of a certain amount of acidic solution in the operation process of this work, the surface of metal particles forms a part of corrosion phenomenon, and eventually forms a pitted irregular spherical surface. If the acidic solution content is reduced or terminated, regular particles can be formed.The Fig. 3 was redraw and uploaded again. And the new discussions of Fig. 2-8 had rework and added in red shown in the revised manuscript.Moreover, according to your comment, the title has revised as "The preparation of FeCo/ZnO composites and enhancement of microwave absorbing property by two-step method". And the grammars have been revised shown in P1L38-40 and P2L46 in the revised manuscript.
Once again, thank you very much for your comments and suggestions. We hope the revisions and responses are sufficient and the resubmitted manuscript is suitable for publication. We shall look forward to hearing from you at your earliest convenience.
Have a good day.
Yours sincerely,
K.GAO, Junliang Zhao, Zhongyi Bai, Wenzheng Song and Rui Zhang
Zhengzhou University of Aeronautics,
Zhengzhou 450015,
R. China
Corresponding authors. E-mail: gaoka9222005@163.com (K. Gao)

Round 2
Reviewer 3 Report
The quality of the figures must be improved, especially XRD, just adding Sample A and Sample B does not improve the resolution of the figure.
What DB card number have you used for the interpretation of the X ray analysis?(From the Rigaku database)
Please present the results obtained in this paper comparatively with the one already existing in the literature, so that you can underline the novelty and the advantages.
L 114 please correct the caption of the Fig. 3 "XRD diagram of FeCo signal-phase signal-phase solid solution and FeCo/ZnO composite"
I already mentioned in my previous review to change "signal-phase" with "single-phase". Please pay attention while modifying your manuscript.
The manuscript has to be revised and edited by a native English speaker.
Author Response
Dear reviewer 3,Re: Manuscript No.: materials-589787
Thank you very much for your reply and help. Thanks a lot for the reviewers’ comments and their kind suggestions of our manuscript entitled “The preparation of FeCo/ZnO composites and enhancement of microwave absorbing property by two-step method”. These comments are all valuable and very helpful for revising and improving our paper, as well as the important guiding significance to our researches. We provide this letter point by point and explain the details of our revisions in the manuscript. In the revised manuscript, we make the revision in red. Besides, we have carefully checked the manuscript and the responses to the reviewers’ comments as follows.
Reviewer #3:
Comment: The quality of the figures must be improved, especially XRD, just adding Sample A and Sample B does not improve the resolution of the figure.
Reply: Thank you for your comment. All the figures with quality have been redrew and uploaded as files in the revised manuscript.
Comment: What DB card number have you used for the interpretation of the X ray analysis?(From the Rigaku database)
Reply: DB card number of FeCo is 01-044-1433. DB card number of ZnO is 01-075-1526.
Comment: Please present the results obtained in this paper comparatively with the one already existing in the literature, so that you can underline the novelty and the advantages.
Reply: Thank you for your comment. We have added the contrasts and explanations in the revised manuscript shown in red to underline the novelty and the advantages.
Comment: L 114 please correct the caption of the Fig. 3 "XRD diagram of FeCo signal-phase signal-phase solid solution and FeCo/ZnO composite"
Reply: Thank you for your comment. This caption of the Fig. 3 has been revised as "Fig.3 XRD diagram of FeCo single-phase solid solution and FeCo/ZnO composite" in red shown in the revised manuscript.
Comment: I already mentioned in my previous review to change "signal-phase" with "single-phase". Please pay attention while modifying your manuscript.
Reply: Thank you for your comment. Sorry for these mistake. "signal-phase" has been revised as "single-phase" in red shown in the revised manuscript
Comment: The manuscript has to be revised and edited by a native English speaker.
Reply: Sorry for English writing. The manuscript has to be revised and edited by a native English speaker.
Once again, thank you very much for your comments and suggestions. We hope the revisions and responses are sufficient and the resubmitted manuscript is suitable for publication. We shall look forward to hearing from you at your earliest convenience.
Have a good day.
Yours sincerely,
K.GAO, Junliang Zhao, Zhongyi Bai, Wenzheng Song and Rui Zhang
Zhengzhou University of Aeronautics,
Zhengzhou 450015,
R. ChinaCorresponding authors. E-mail: gaoka9222005@163.com (K. Gao)

Reviewer 4 Report
The authors appear to have addressed the comments suggested to improve their manuscript.
Author Response
Thank you for your comment and your useful help.

Round 3
Reviewer 3 Report
I did not find in the manuscript the results obtained in this paper comparatively with the one already existing in the literature, so that you can underline the novelty and the advantages. For this use at least five significant studies in the field.(present the comparison in a table)
The conclusions presents results and to many details, please rewrite.
I recommend you to use MDPI English Editing Service.
Author Response
Dear reviewer 3,Re: Manuscript No.: materials-589787
Thank you very much for your reply and help. Thanks a lot for the reviewers’ comments and their kind suggestions of our manuscript entitled “The preparation of FeCo/ZnO composites and enhancement of microwave absorbing property by two-step method”. These comments are all valuable and very helpful for revising and improving our paper, as well as the important guiding significance to our researches. We provide this letter point by point and explain the details of our revisions in the manuscript. In the revised manuscript, we make the revision in red. Besides, we have carefully checked the manuscript and the responses to the reviewers’ comments as follows.
Reviewer #3:
Comment: I did not find in the manuscript the results obtained in this paper comparatively with the one already existing in the literature, so that you can underline the novelty and the advantages. For this use at least five significant studies in the field.(present the comparison in a table)
The conclusions presents results and to many details, please rewrite.
I recommend you to use MDPI English Editing Service.
Reply: Thank you for your comment. according to your comment, we have added the Table 2 including the comparisons with the one already existing in the literature to underline the novelty and the advantages of our work. This was in the revised manuscript shown in red
The conclusions has been rewritten shown in red in the revised manuscript.
And we have checked English writing of this paper. The manuscript has to be revised and edited by a native English speaker.
Once again, thank you very much for your comments and suggestions. We hope the revisions and responses are sufficient and the resubmitted manuscript is suitable for publication. We shall look forward to hearing from you at your earliest convenience.
Have a good day.
Yours sincerely,
K.GAO, Junliang Zhao, Zhongyi Bai, Wenzheng Song and Rui Zhang
Zhengzhou University of Aeronautics,
Zhengzhou 450015,R. China
Corresponding authors. E-mail: gaoka9222005@163.com (K. Gao)

Reviewer 4 Report
The paper can be accepted in this form.
Author Response
Thank your for your useful comments.